# Low First Trimester Pregnancy-Associated Plasma Protein-A Levels Are Not Associated with an Increased Risk of Intrapartum Fetal Compromise or Adverse Neonatal Outcomes: A Retrospective Cohort Study

**DOI:** 10.3390/jcm9041108

**Published:** 2020-04-13

**Authors:** Jessica M. Turner, Sailesh Kumar

**Affiliations:** 1Mater Research Institute, University of Queensland, Raymond Terrace, Level 3 Aubigny Place, S. Brisbane, Queensland 4101, Australia; jessica.turner@uq.edu.au; 2Faculty of Medicine, University of Medicine, Whitty Building, Annerley Road, S. Brisbane, Queensland 4101, Australia

**Keywords:** fetal compromise, PAPP-A, emergency cesarean birth, adverse perinatal outcomes, biochemical markers, biomarker

## Abstract

The aim of this study was to assess if women with a low first trimester maternal pregnancy-associated plasma protein-A (PAPP-A) level are at increased risk of emergency cesarean (EmCS) for intrapartum fetal compromise (IFC) and/or adverse neonatal outcomes. This was a retrospective cohort study performed at Mater Mother’s Hospital, Brisbane, Australia, between 2016 and 2018. All women with a singleton, euploid, non-anomalous fetus with a documented PAPP-A level measured between 10 ^+0^ and 13 ^+6^ weeks gestation during the study period were included. Data were extracted from the institution’s perinatal database and dichotomized according to PAPP-A level (≤0.4 Multiples of Medium (MoM) vs. >0.4 MoM). The primary outcomes were EmCS-IFC and a composite of severe adverse neonatal outcomes (SCNO). Nine thousand sixty-one pregnancies were included, 3.3% with a PAPP-A ≤ 0.4 MoM. Low maternal PAPP-A was not associated with an increased risk of EmCS-IFC (adjusted odds ratio (aOR) 0.77, 95% confidence interval (CI) 0.24–2.46, *p* = 0.66) or SCNO (aOR 0.65, 95% CI 0.39–1.07, *p* = 0.09). Low PAPP-A was associated with increased odds of pre-eclampsia, preterm birth and birthweight < 10th centile. In conclusion, low maternal PAPP-A level is not associated with an increased risk of EmCS IFC or adverse neonatal outcomes despite greater odds of low-birthweight infants and preterm birth.

## 1. Introduction

Fetal hypoxia in labor occurs when there is a mismatch between utero-placental perfusion and fetal oxygen and metabolic requirements in between uterine contractions [1,2]. Whilst most fetuses can tolerate the reduction in placental perfusion during labor, some cannot and are thus at risk of hypoxic injury [3]. Indeed, perinatal hypoxia is a major contributor to a myriad of adverse outcomes, including stillbirth, neonatal death, hypoxic ischemic encephalopathy and severe neonatal morbidity. Almost 10% of the global burden of disease is attributable to complications arising in the neonatal period [4,5]. Pregnancies that are complicated by pre-existing placental dysfunction, regardless of the size of the fetus, are at increased risk of perinatal complications [3,6,7,8]. Therefore, identification of pre-existing placental dysfunction is the focus of considerable research effort. 

Pregnancy-associated plasma protein-A (PAPP-A) is a placental glycoprotein produced by the syncytiotrophoblast [9,10]. It plays a critical role in regulating the availability of insulin-like growth factor (IGF) through the proteolysis of IGF-binding protein-4 and is a key factor in modulating placental development and subsequent fetal growth [11]. It is a component of the combined first trimester aneuploidy screening test and is routinely measured between 10 ^+0^ and 13 ^+6^ weeks gestation (11 ^+2^ to 14 ^+1^ weeks in some institutions) [12,13,14]. Low first trimester levels of maternal PAPP-A are associated with an increased risk of pre-eclampsia (PE), fetal growth restriction (FGR) and preterm birth (PTB), all conditions with a significant placental etiology [15,16,17,18]. Although there are varying definitions of what constitutes a low PAPP-A level, most investigators now recognize that levels below the 5th centile (≤0.4 Multiples of the Medium (MoM)) for gestation are a significant risk factor for adverse obstetric and perinatal outcomes [16,19,20,21].

Given the association between maternal PAPP-A levels and placental function, we investigated the correlation between first trimester PAPP-A levels and the risk of emergency cesarean section (EmCS) for intrapartum fetal compromise (IFC) (“fetal distress”) and adverse neonatal outcomes. We hypothesized that low PAPP-A levels would be associated with an increased risk of EmCS for IFC and adverse neonatal outcomes.

## 2. Materials and Methods

This was a retrospective cohort study of 9061 women at the Mater Mothers’ Hospital in Brisbane, Australia, between January 2016 and September 2018. Ethical, governance and privacy approvals were obtained from the Mater Human Research Ethics Committee and Governance office respectively (Ref no. HREC/18/MHS/46) on 1st May 2018.

All women with a singleton, euploid, non-anomalous (no known congenital anomaly) fetus with a documented first trimester PAPP-A level and birth > 23 weeks gestation [22,23] were eligible for inclusion. Gestational age was calculated from a first trimester ultrasound scan. Over the study period, PAPP-A levels were measured using two different assay platforms (Delfia^®^ Xpress and BRAHMS^TM^ Kryptor). PAPP-A MoM was corrected for weight, ethnicity, smoking status, parity and gestation.

In our institution, low PAPP-A was defined as ≤5th centile (≤0.4 MoM or >0.4 MoM). Women with a low PAPP-A in our institution undergo additional surveillance with third trimester fetal growth ultrasound scans at 28 and 34 weeks of gestation.

Primary outcomes were EmCS for IFC and a composite of severe adverse neonatal outcomes (SCNO) (5 min Apgar ≤3, severe acidosis, neonatal intensive care unit (NICU) admission or perinatal death).

Secondary outcomes (obstetric and neonatal) included mode of birth (spontaneous vaginal birth (SVB), instrumental, planned CS, EmCS), preterm birth (PTB) (<37 weeks of gestation) and postpartum hemorrhage (PPH) (blood loss ≥ 500 mL). Fetal compromise in labor was defined as a pathological fetal heart rate (FHR) pattern as determined by the treating obstetric team according to guidelines from the Royal Australia and New Zealand College of Obstetricians and Gynecologists [24] (prolonged bradycardia (<100 bpm for >5 min)/absent baseline variability, sinusoidal pattern, complicated variable decelerations with reduced or absent baseline variability or late decelerations with reduced or absent baseline variability). Neonatal outcomes were birthweight (BW), small for gestational age (SGA) infant (BW < 10th centile for gestational age and gender [25]), 5 min Apgar score <7 and ≤3, severe acidosis (umbilical artery pH < 7.0 or base excess ≤12.0 mmol/L), NICU admission and perinatal death (stillbirth or neonatal death within 28 days of birth). Data were dichotomized according to PAPP-A levels (≤0.4 MoM or >0.4 MoM).

## 3. Statistical Analysis

Data were reported as mean (standard deviation (SD)) or median (interquartile range (IQR)), and associations between variables were assessed using Pearson’s χ^2^, Fischer’s exact test, Student’s *t*-test or Wilcoxon Rank-sum test as appropriate. Generalized estimating equations were used to account for mothers who gave birth more than once during the study period. Multiple logistic regression analyses were performed, adjusting for hypertension, gestation at birth and BW where indicated. Two subgroup analyses were also performed: (1) women birthing <37 weeks of gestation and (2) women with infants that were SGA. Given the varying definitions utilized in the literature of low PAPP-A, a post-hoc analysis was performed to assess whether different PAPP-A MoM thresholds affected the outcomes of interest; *p* values ≤ 0.05 were considered significant. Statistical analysis was performed using Stata SE^®^, Release 15 (StataCorp LP, College Station, TX, USA).

## 4. Results

PAPP-A levels were available for 9602 (34.2%) women delivering at Mater Mothers’ Hospital, Brisbane between January 2016 and September 2018, of whom 9061 (94.4%) met the inclusion criteria. Of these, 301 women (3.3%) had a PAPP-A level ≤ 0.4 MoM, and 8760 had a PAPP-A level >0.4 MoM for gestation (Figure 1).

Women with low PAPP-A levels had marginally higher BMI (23.6 kg/m^2^ (21.0–27.0) vs. 23.0 kg/m^2^ (20.6–26.5), *p* = 0.04), lower Socio-Economic Index For Area score (1012.2 (75.4) vs. 1023.2 (68.1), *p* = 0.03) and higher rates of hypertension (8.0% vs. 3.9%, *p* < 0.001) (Table 1).

Low PAPP-A levels did not increase the odds of EmCS IFC (1.0% vs. 1.3%; aOR 0.77, 95% CI 0.24–2.46, *p* = 0.66). It was however associated with significantly decreased odds of achieving a SVB or any vaginal birth (SVB and instrumental) (SVB 49.2% vs. 55.3%; aOR 0.79, 95% CI 0.63–0.99, *p* = 0.045 and vaginal birth 63.8% vs. 70.5%; aOR 0.75, 95% CI 0.59–0.96, *p* = 0.02) and higher odds of planned CS (17.3% vs. 12.1%; aOR 1.46, 95% CI 1.07–1.98, *p* = 0.02) (Table 2).

Women with low levels were more likely to deliver preterm (13.0% vs. 5.3%, aOR 2.51, 95% CI 1.76–3.56, *p* < 0.001; positive likelihood ratio (PLR) 2.53) with a significant difference noted for those delivering both early and late preterm but not at extreme preterm gestations (32 + 0–36 + 6 weeks: 10.6% vs. 4.5%; aOR 2.41, 95% CI 1.65–3.52; 28 + 0–31 + 6 weeks: 2.0% vs. 0.5%; aOR 3.59, 95% CI 1.47–8.75, *p* < 0.001; <28 + 0 weeks: 0.3% vs. 0.3%; aOR 0.91, 95% CI 0.12–6.89, *p* = 0.93). Additionally, women with low PAPP-A levels had higher odds of early term birth (39.2% vs. 28.7%; aOR 1.56, 95% CI 1.23–1.99, *p* < 0.001) (Table 2).

Low PAPP-A also increased the odds of an SGA infant (BW < 10th centile: 22.7% vs. 9.1%; aOR 2.93, 95% CI 2.22–3.88, *p* < 0.001; PLR 2.80; birthweight <5th centile: 14.3% vs. 4.0%; aOR 3.97, 95% CI 2.83–5.60, *p* < 0.001; PLR 3.63). There were no differences for any of the other adverse neonatal outcomes, including SCNO (6.3% vs. 5.8%; aOR 0.65, 95% CI 0.39–1.07, *p* = 0.09) (Table 3).

Subgroup analysis of 503 women birthing <37 weeks of gestation demonstrated no difference in odds for EmCS for IFC for those with a low PAPP-A level (5.1% vs. 2.2%; aOR 2.35, 95% CI 0.46–12.04, *p* = 0.30). The association between low PAPP-A and low birthweight persisted in this cohort (birthweight < 10th centile: 30.8% vs. 10.0%; aOR 3.72, 95% CI 1.73–8.04, *p* = 0.001; birthweight <5th centile: 18.0% vs. 5.0%; aOR 4.03, 95% CI 1.56–10.36, *p* = 0.004). There was no significant difference in the other outcomes amongst this cohort of preterm deliveries (Table 4).

Subgroup analysis of 869 women who delivered a baby with a BW < 10th centile again demonstrated no difference in the odds for EmCS for IFC (1.5% vs. 1.9%; aOR 0.80, 95% CI 0.10–6.20, *p* = 0.83). Women in this cohort with a low PAPP-A were more likely to deliver preterm (17.4% vs. 5.8%; aOR 3.25, 95% CI 1.57–6.70, *p* = 0.001) and deliver an infant with a BW < 5th centile for gestational age (62.3% vs. 44.3%; aOR 2.14, 95% CI 1.29–3.54, *p* = 0.003). There was no difference in other outcomes in this cohort (Appendix A).

## 5. Discussion

The results of this large study demonstrate that a low first trimester PAPP-A level is not associated with an increase in the odds of EmCS for IFC or adverse neonatal outcomes despite being a strong risk factor for low BW and PTB. In common with other studies [26], we found that low maternal PAPP-A levels were associated with reduced odds of achieving an SVB or any vaginal birth. Subgroup analyses of women birthing preterm or those with an SGA infant failed to demonstrate any difference in outcomes according to maternal PAPP-A levels.

Our results concur with a small prospective study by Livrinova et al. that showed a lack of difference in rates of EmCS in women with PAPP-A levels ≤ 0.4 MoM [27]. Although this study also reported “fetal hypoxia” as an outcome, this was not defined, making interpretation somewhat difficult. However, contrary to our findings, two much smaller studies by Avşar et al. (283 women; low PAPP-A defined as ≤ 0.5 MoM) [28] and Uccella et al. (a prospective cohort study of 1037 women; low PAPP-A defined as < 0.52 MoM) [29] reported higher rates of emergency CS for IFC in the low PAPP-A cohort. Possible reasons for the discrepancy in findings include the lack of blinding of PAPP-A levels in the study by Uccella et al. Planned CS rates were higher in women with low PAPP-A levels—possible explanations for this include higher rates of planned CS for fetal malpresentation (23.1 vs. 10.2%). The reasons for this are unclear from either our data or the published literature. Low PAPP-A is not an indication for planned CS in our institution. PAPP-A increases the bioavailability and half-life of IGF [30] and plays a critical role in modulating trophoblast growth, differentiation and invasion in a temporal spatial manner [31,32]. Low PAPP-A levels are associated with small and dysfunctional placentae with histological evidence of inadequate endovascular trophoblast invasion of the maternal spiral arteries [33]. Furthermore, women with placentae showing histopathological evidence of underperfusion and fetal vascular lesions have significantly lower first trimester PAPP-A levels than do controls [34].

Some of the placental abnormalities (high resistance, low flow circulation predisposing to hypo-perfusion, hypoxia, re-perfusion injury and oxidative stress [35]) associated with low maternal PAPP-A levels could render the fetus vulnerable to intrapartum compromise through several mechanisms [3]. A greater reduction in uteroplacental perfusion (and consequently a greater degree of fetal hypoxia) occurs for any given increase in intrauterine pressure due to the smaller surface area of the unconverted spiral arteries and sustained vasoreactivity [36,37,38]. Additionally, the ability of an SGA fetus to transition to anaerobic metabolism in response to intrapartum hypoxia is compromised because of lower fetal glycogen stores [39] with even short periods of hypoxia quickly compromising cerebral and cardiac glycogen levels by up to 80% [40].

Although our hypothesis that low PAPP-A levels, through their effect on abnormal placental development and function, would be associated with an increased risk of intrapartum fetal compromise was biologically plausible, our results failed to demonstrate this. This may be due to several reasons—while PAPP-A plays a crucial role in placental development, it is one of many modulators, and as such, analysis of its association with poor placental function independent of other markers may be an oversimplification of the highly complex processes that occur during placentation.

## 6. Limitations

There are several limitations with our study, including those inherent to its retrospective nature and its tertiary center focus with a significant load of high-risk pregnancies, which may affect the generalizability of our results. In our institution, low PAPP-A is an indication for fetal growth surveillance with third trimester ultrasound scans. This may limit the generalizability of our results to institutions where this is not policy. However, PAPP-A does not influence the mode of birth decisions in our institution; in particular, it is not policy to recommend early term delivery or planned CS among those with a low PAPP-A. Additionally, PAPP-A levels in our study population were measured at different laboratories using two different assay platforms (Delfia^®^ Xpress, PerkinElmer, Waltham, MA, USA and BRAHMS^TM^ Kryptor, Thermo Fisher Scientific Inc., Waltham, MA, USA)—this lack of standardization may have influenced our results. Another limitation is the lack of evaluation of other biochemical or physical markers of placentation and/or maternal hemodynamics in the third trimester, closer to the delivery or correlation with placental histopathology. 

Finally, in an era where aneuploidy screening is rapidly evolving to a genomic-based approach using non-invasive prenatal testing systems, it is likely that maternal PAPP-A levels will soon no longer be available, as it is unlikely to be measured in the first trimester outside of the first trimester combined screening test, particularly in high-income countries. We acknowledge that this transition will preclude the clinical relevance of our findings. However, our results may still be pertinent in countries or healthcare systems that continue to use the current model of first trimester combined testing.

## 7. Conclusions

This large study demonstrates that low first trimester maternal PAPP-A levels are not associated with an increased risk of operative birth for IFC nor adverse neonatal outcomes despite increased odds of PTB and SGA.

## Figures and Tables

**Figure 1 jcm-09-01108-f001:**
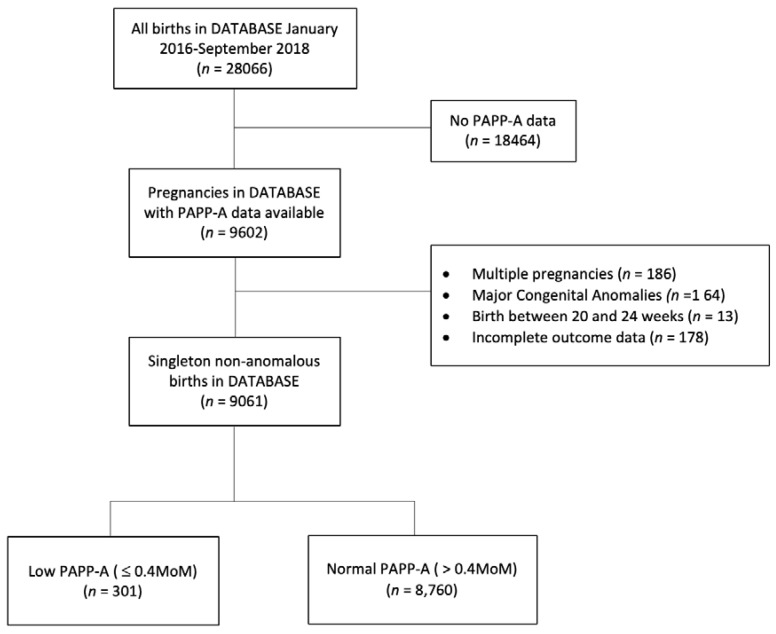
Study flow diagram. PAPP-A, pregnancy-associated plasma protein-A.

**Table 1 jcm-09-01108-t001:** Maternal Demographics.

	PAPP-A ≤ 0.4 MoM	PAPP-A > 0.4 MoM	*p* Value
*n*	301	8760	
PAPP-A MoM (median(IQR))	0.34 (0.29–0.37)	1.18 (0.83–1.69)	
Age (mean(sd))	31.5 (4.8)	31.1 (4.8)	0.16
BMI (median(IQR))	23.6 (21.0–27.0)	23.0 (20.6–26.5)	0.04
**Ethnicity**			
Caucasian	48.8 (147)	50.6 (4428)	0.56
ATSI	2.3 (7)	1.5 (130)	0.24
Asian	19.9 (60)	23.5 (2059)	0.15
Other	9.0(27)	6.3 (551)	0.06
Nulliparous	45.5 (137)	49.1 (4300)	0.22
Previous cesarean section	18.9 (57)	14.9 (1309)	0.06
Trial of labor after CS	28.1 (16)	34.7 (454)	0.30
Diabetes	15.3 (46)	13.1 (1144)	0.26
Hypertension (current)	8.0 (24)	3.9 (342)	< 0.001
Pre-existing hypertension	1.1 (3)	0.7 (62)	0.003
Gestational hypertension	0.4 (1)	0.3 (22)	
Pre-eclampsia	3.5 (10)	1.1 (94)	
Smoker	5.0 (15)	3.9 (337)	0.32
Induction of labor	16.6 (50)	16.8 (1470)	0.94

Data presented as % (*n*), mean (sd) or median (IQR) as appropriate. PAPP-A MoM, pregnancy-associated plasma protein-A multiples of the median; BMI, body mass index kg/m^2^; ATSI, Aboriginal Torres Strait Islander; CS, cesarean section.

**Table 2 jcm-09-01108-t002:** Intrapartum outcomes.

	PAPP-A ≤ 0.4 MoM	PAPP-A > 0.4 MoM	*p*	Unadjusted OR (95% CI)	Adjusted OR (95% CI)	*p*
*n*	301	8760				
**Mode of Birth**						
Spontaneous vaginal *	49.2 (148)	55.3 (4843)	0.04	0.78 (0.62–0.98)	0.79 (0.63–0.99)	0.045
Instrumental *	14.6 (44)	15.3 (1336)	0.76	0.95 (0.69–1.32)	0.98 (0.70–1.35)	0.88
All vaginal births *	63.8 (192)	70.5 (6179)	0.01	0.74 (0.58–0.93)	0.75 (0.59–0.96)	0.02
Planned CS *	17.3 (52)	12.1 (1062)	0.007	1.51 (1.11–2.06)	1.46 (1.07–1.98)	0.02
Emergency CS *	18.9 (57)	17.3 (1519)	0.47	1.11 (0.83–1.49)	1.10 (0.82–1.48)	0.52
*EmCS intrapartum fetal compromise*	5.3 (3)	7.3 (111)	0.56	0.70 (0.22–2.29)	0.71 (0.22–2.29)	0.56
Gestation at birth (weeks)	38 (37–39)	39 (38–40)	<0.001	-	-	-
Extreme preterm birth (< 28 weeks) *	0.3 (1)	0.3 (29)	1.00	1.00 (0.14–7.39)	0.91 (0.12–6.89)	0.93
Early preterm birth (28.0–31.6 weeks) *	2.0 (6)	0.5 (43)	<0.001	4.12 (1.74–9.76)	3.59 (1.47–8.75)	<0.001
Late preterm birth (32.0–36.6 weeks) *	10.6 (32)	4.5 (392)	<0.001	1.54 (1.72–3.75)	2.41 (1.65–3.52)	<0.001
Early Term (37.0–38.6 weeks) *	39.2 (118)	28.7 (2513)	<0.001	1.60 (1.26–2.03)	1.56 (1.23–1.99)	<0.001
Term (39.0–40.6 weeks) *	43.2 (130)	53.6 (4691)	<0.001	0.66 (0.52–0.83)	0.68 (0.54–0.86)	0.001
Late Term (41.0–41.6 weeks) *	4.7 (14)	12.2 (1069)	<0.001	0.35 (0.20–0.60)	0.36 (0.21–0.61)	<0.001
Post Term (≥ 42.0 weeks) *	0	0.3 (23)	1.00	n/a	-	-
**Preterm Birth**						
Spontaneous Preterm birth	79.5 (31)	81.7 (379)	0.74	0.87 (0.38–1.97)	0.88 (0.37–2.14)	0.78
Medically indicated preterm birth	20.5 (8)	18.3 (85)		1.15 (0.51–2.61)	1.13 (0.47–2.74)	0.78

Data presented as % (*n*), odds ratio (95% confidence interval); PAPP-A MoM, pregnancy-associated plasma protein-A multiples of the median; OR, odds ratio; CI, confidence interval; CS, cesarean section; IFC, intrapartum fetal compromise. * adjusted for hypertension.

**Table 3 jcm-09-01108-t003:** Perinatal outcomes.

	PAPP-A ≤ 0.4 MoM	PAPP-A > 0.4 MoM	*p*	Unadjusted OR (95% CI)	Adjusted OR (95% CI)	*p*
*n*	301	8760				
Birthweight	3037.8 (630.5)	3365.8 (5253.6)	<0.001	-	-	-
Birthweight < 10th centile *	22.9 (69)	9.1 (800)	<0.001	2.96 (2.24–3.91)	2.93 (2.22–3.88)	< 0.001
Birthweight < 5th centile *	14.3 (43)	4.0 (354)	<0.001	3.96 (2.81–5.56)	3.97 (2.83–5.60)	< 0.001
5 min Apgar < 7 ^	1.7 (5)	2.0 (178)	0.84	0.81 (0.33–1.99)	0.45 (0.17–1.15)	0.10
5 min Apgar < 3 ^	0	0.3 (24)	1.00	-	-	-
Acidosis ^	0.3 (1)	0.4 (35)	1.00	0.83 (0.11–6.09)	0.68 (0.09–4.99)	0.71
NICU admission ^	6.3 (19)	5.3 (462)	0.43	1.21 (0.75–1.94)	0.77 (0.47–1.26)	0.30
Respiratory distress syndrome ^	17.9 (54)	17.1 (1497)	0.71	1.06 (0.78–1.44)	0.81 (0.60–1.08)	0.15
Perinatal death ^	0.7 (2)	0.4 (37)	0.37	1.58 (0.38–6.57)	0.75 (0.18–3.17)	0.70
Intrauterine fetal demise	0	0.3 (24)	1.00	n/a	-	-
Neonatal death ^	0.7 (2)	0.2 (13)	0.09	4.50 (1.01–20.04)	2.20 (0.45–10.7)	0.33
SCNO ^	6.3 (19)	5.8 (508)	0.71	1.09 (0.68–1.75)	0.65 (0.39–1.07)	0.09

Data presented as % (*n*), odds ratio (95% confidence interval); PAPP-A MoM, pregnancy-associated plasma-protein-A multiples of the median; OR, odds ratio; CI, confidence interval; Acidosis, umbilical cord pH < 7.0 or base excess ≤ −12.0; NICU, neonatal intensive care unit; perinatal death, still birth and neonatal death; SCNO = acidosis, NICU admission, 5 min Apgar < 3, perinatal death. * adjusted for hypertension. ^ adjusted for hypertension, birthweight and gestation.

**Table 4 jcm-09-01108-t004:** Outcomes amongst those delivering prior to 37 weeks of gestation.

	PAPP-A ≤ 0.4 MoM	PAPP-A > 0.4 MoM	*p*	Adjusted OR (95% CI)	*P*
*n*	39	464			
Pre-eclampsia	2.6 (1)	1.7 (8)	0.70	1.5 (0.18–12.34)	0.71
**Mode of Birth**					
All vaginal births *	48.7 (19)	54.7 (254)	0.51	0.86 (0.43–1.75)	0.69
Planned CS *	10.3 (4)	9.7 (45)	0.78	1.08 (0.37–3.12)	0.89
Emergency CS *	41.0 (16)	35.6 (165)	0.49	1.13 (0.53–2.40)	0.75
*Emergency CS IFC* *	5.1 (2)	2.2 (10)	0.24	2.35 (0.46–12.04)	0.30
Gestation at birth (weeks)	35 (33–36)	35 (34–36)	0.18	-	-
Birthweight	2108.4 (662.0)	2372.7 (738.5)	0.03	-	-
Birthweight < 10th centile *	30.8 (12)	10.0 (46)	< 0.001	3.72 (1.73–8.04)	0.001
SCNO ^	28.2 (11)	31.5 (146)	0.67	0.67 (0.30–1.51)	0.34

Data presented as % (*n*), odds ratio (95% confidence interval); PAPP-A MoM, pregnancy-associated plasma protein-A multiples of the median; OR, odds ratio; CI, confidence interval; CS, cesarean section; IFC, intrapartum fetal compromise; SCNO = acidosis, NICU admission, 5 min Apgar < 3, perinatal death. * adjusted for hypertension. ^ adjusted for hypertension, gestation and birthweight.

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
