# Peer review of "Low First Trimester Pregnancy-Associated Plasma Protein-A Levels Are Not Associated with an Increased Risk of Intrapartum Fetal Compromise or Adverse Neonatal Outcomes: A Retrospective Cohort Study"

_jcm, 2020, doi:10.3390/jcm9041108_

Round 1

Reviewer 1 Report

Thank you for giving me the opportunity to review your paper. Below are some suggested revisions:

Page 2 Line 54-Consider revision/inclusion.  Although NICE guidance suggests the upper limit for first trimester screening is 13+6, the UKSNSC FASP programme includes testing until 14+1

Page 2 - line 71- please define non-anomalous fetus

Page 2 Line- 75- please confirm that the PAPP-A MoMs used are corrected for weight, ethnicity, smoking and other effects.

Page 4- Figure 1 flow diagram- Please confirm the figures/consider reivison. Figure 1  seems to suggest that there were no congenital anomalies and only 2 miscarriages out of 9219 pregnancies. This seems unlikely. Also there is no mention of the number of pregnancies excluded due to aneuploidy

Page 5- Table 1-Please confirm  if pre-eclampsia relates to this pregnancy or previous history of pre-eclampsia.

Page 5- Line 133- Please add the data for the extreme preterm gestations

Line 143- Table 2 Please adjust column heading cells as all text not visible

Page 8- Table 3- Please adjust formatiing headers and p values not  fully visible

Page  9 Line  182-You mention that in another study there was no blinding to PAPPA results.  Your paper needs to clarify whether women underwent any additional testing or entered a different maternity pathway on the basis of the low PAPPA from the aneuploidy screen and how those interventions may have impacted the results.

Page 9 Line 184- There are differences in PAPPA levels in differences in ethnic groups, however these should be accounted for upon production of a MoM value from the screening lab which should correct for these factors. Therefore this statement is not valid unless the MoMs have not been corrected for ethnicity. This requires clarification.

Page 9 Line 187- Please confirm the higher planned CS rates were not part of normal interventions due to low PAPPA

General comments-

This paper provides some evidence that low levels of PAPPA are not linked to an increase in the likelihood of emergency section for intrapartum fetal compromise but does show low PAPPA is linked to SGA, and preterm birth which is already very well documented in the literature.   The paper needs to  be strengthened by addressing a couple of issues. The first is related to the PAPPA MoMs obtained. It needs to be clarified that the MoM  is corrected for the many factors that can effect PAPP-A levels (particularly ethnicity, weight and smoking), especially as the authors allude to ethnicity as a cause of higher PAPP-A results in their discussion.

Secondly the authors need to clarify whether women who received a low PAPPA result were treated any differently for the rest of their antepartum care compared to the control group.  If so this bias needs to be addressed particularly how it may have changed decision making regarding delivery in those women.  

The discussion  doesn't state what this study adds to the knowledge base or what is novel about it. This needs to be added.

Author Response

Page 2 Line 54-Consider revision/inclusion.  Although NICE guidance suggests the upper limit for first trimester screening is 13+6, the UKSNSC FASP programme includes testing until 14+1. The following has been added to this sentence: (11+2 to 14+1 weeks in some institutions). NB the reference will need to be formatted as we were unable to correctly insert the reference citation.

Page 2 - line 71- please define non-anomalous fetus – have amended to define as “no known congenital anomaly”

Page 2 Line- 75- please confirm that the PAPP-A MoMs used are corrected for weight, ethnicity, smoking and other effects. PAPP-A MoMs are corrected for weight, ethnicity, smoking status, parity and gestation. Manuscript amended.

Page 4- Figure 1 flow diagram- Please confirm the figures/consider reivison. Figure 1  seems to suggest that there were no congenital anomalies and only 2 miscarriages out of 9219 pregnancies. This seems unlikely. Also there is no mention of the number of pregnancies excluded due to aneuploidy – this has been amended

Page 5- Table 1-Please confirm if pre-eclampsia relates to this pregnancy or previous history of pre-eclampsia. – this relates to current pregnancy. Have clarified this in Table 1.

Page 5- Line 133- Please add the data for the extreme preterm gestations – manuscript amended

Line 143- Table 2 Please adjust column heading cells as all text not visible – have amended

Page 8- Table 3- Please adjust formating headers and p values not  fully visible - have amended

Page  9 Line  182-You mention that in another study there was no blinding to PAPPA results.  Your paper needs to clarify whether women underwent any additional testing or entered a different maternity pathway on the basis of the low PAPPA from the aneuploidy screen and how those interventions may have impacted the results. Added to manuscript page 3, line 78: Women with a low PAPP-A in our institution undergo additional surveillance with third trimester fetal growth ultrasound scans at 28 and 34 weeks gestation.

Page 9 Line 184- There are differences in PAPPA levels in differences in ethnic groups, however these should be accounted for upon production of a MoM value from the screening lab which should correct for these factors. Therefore this statement is not valid unless the MoMs have not been corrected for ethnicity. This requires clarification. Agree with this and as such we have removed this statement from the discussion.

Page 9 Line 187- Please confirm the higher planned CS rates were not part of normal interventions due to low PAPPA – following sentence inserted into manuscript: Low PAPP-A is not an indication for planned CS in our institution.

General comments-

This paper provides some evidence that low levels of PAPPA are not linked to an increase in the likelihood of emergency section for intrapartum fetal compromise but does show low PAPPA is linked to SGA, and preterm birth which is already very well documented in the literature.   The paper needs to be strengthened by addressing a couple of issues. The first is related to the PAPPA MoMs obtained. It needs to be clarified that the MoM  is corrected for the many factors that can effect PAPP-A levels (particularly ethnicity, weight and smoking), especially as the authors allude to ethnicity as a cause of higher PAPP-A results in their discussion.

Secondly the authors need to clarify whether women who received a low PAPPA result were treated any differently for the rest of their antepartum care compared to the control group.  If so this bias needs to be addressed particularly how it may have changed decision making regarding delivery in those women. Added to limitations in discussion: - In our institution low PAPP-A is an indication for fetal growth surveillance with third trimester ultrasound scans. This may limit the generalizability of our results to institutions where this is not policy. However, PAPP-A does not influence mode of birth decisions in our institution, in particular it is not policy to recommend early term delivery or planned CS amongst those with a low PAPP-A.

The discussion doesn't state what this study adds to the knowledge base or what is novel about it. This needs to be added. – We have clearly stated that our study is one of the largest on this topic and conclusively demonstrates that despite the association with preterm birth and small for gestational age, low maternal PAPP-A is not a risk factor for intrapartum fetal compromise.

Reviewer 2 Report

Rather minor points:

  1. For Table 1, I found it a bit of an effort to look at which results were in either median (IQR) or mean (SD). May I suggest having under each main variable, which are in bold, highlighting the units i.e. Hypertension, n (%) or BMI, median (IQR). You could also use an asterisk for those that are mean (SD) and another symbol for the others.
  2. You mention under the discussion starting line 181, that ethnicity may play a role in influencing PAPP-A levels. Was this considered in any of the analyses? 
  3. Under references, line 370, the reference has been split. Can you correct to flow properly?

Author Response

Rather minor points:

  1. For Table 1, I found it a bit of an effort to look at which results were in either median (IQR) or mean (SD). May I suggest having under each main variable, which are in bold, highlighting the units i.e. Hypertension, n (%) or BMI, median (IQR). You could also use an asterisk for those that are mean (SD) and another symbol for the others. – have clarified this in table 1
  2. You mention under the discussion starting line 181, that ethnicity may play a role in influencing PAPP-A levels. Was this considered in any of the analyses? – this has been amended following corrections as recommended by reviewer 1 and this part of the discussion was removed as PAPP-A MoM adjust for ethnicity so this was no longer an issue.
  3. Under references, line 370, the reference has been split. Can you correct to flow properly? - corrected

Reviewer 3 Report

This is a retrospective study that evaluated the association between low first-trimester PAPP-A and pregnancy outcomes. The authors observed no significant changes in their primary outcome of interest which include emergency caesarean (EmCS) for intrapartum fetal compromise (IFC) and/or adverse neonatal outcomes. They observed no significant association.

The strenght is the large sample size. Statistics are appropriate and odds ratio are adjusted for birthweight and gestational age. While it can be appropriate to make such adjustment, it is tricky because you are adjusting for co-morbidities (preterm birth and IUGR).

Finally, low PAPP-A has been defined as <0.4 MoM, which is an accepted definition, but other cut-off have been used. It could have been interesting to see whether a dose-response effect could be present or if other cut-off could have lead to significant associations.

I would strongly recommand the following adds to the authors:

1) In the abstract, please mention that PTB, IUGR and PREECLAMPSIA is increased with low PAPP-A. These significant associations cannot be overlooked.

2) Please add a table (not in the supplemental) for this specific adverse pregnancy outcomes : preterm preeclampsia, preterm IUGR, preterm SNCO, with their appropriate OR and aOR

Author Response

1) In the abstract, please mention that PTB, IUGR and PREECLAMPSIA is increased with low PAPP-A. These significant associations cannot be overlooked. – this has been amended

2) Please add a table (not in the supplemental) for this specific adverse pregnancy outcomes : preterm preeclampsia, preterm IUGR, preterm SNCO, with their appropriate OR and aOR – Table 4 has been added to the manuscript containing this information and therefore Table S1 can be removed. Table S2 now titled Table S1.